# CheXT: Knowledge-Guided Cross-Attention Transformer for Abnormality Classification and Localization in Chest X-rays

## Abstract

Classical chest X-ray analysis has designed radiomic features to indicate the characteristics of abnormality of the chest X-rays. However, extracting reliable radiomic features heavily hinges on pathology localization, which is often absent in real-world image data. Although the past decade has witnessed the promising performance of convolutional neural networks (CNNs) in analyzing chest X-rays, most of them ignored domain knowledge such as radiomics. Recently, the surge of Transformers in computer vision has suggested a promising substitute for CNNs. It can encode highly expressive and generalizable representations and avoid costly manual annotations via a unique implementation of the self-attention mechanism. Moreover, Transformers naturally suit the feature extraction and fusion from different input modalities. Inspired by its recent success, this paper proposes **CheXT**, the first Transformer-based chest X-ray model. CheXT targets (semi-supervised) abnormality classification and localization from chest X-rays, enhanced by baked-in auxiliary knowledge guidance using radiomics. Specifically, CheXT consists of an image branch and a radiomics branch, interacted by cross-attention layers. During training, the image branch leverages its learned attention to estimate pathology localization, which is then utilized to extract radiomic features from images in the radiomics branch. Therefore, the two branches in CheXT are deeply fused and constitute an end-to-end optimization loop that can bootstrap accurate pathology localization from image data without any bounding box used for training. Extensive experiments on the NIH chest X-ray dataset demonstrate that CheXT significantly outperforms existing baselines in disease classification (by 1.1% in average AUCs) and localization (by a **significant average margin of 3.6%** over different IoU thresholds). Codes and models will be publicly released.

## 1 Introduction

In medical study, *handcrafted radiomics* (Zwanenburg et al., 2016) refers to the process of extracting several quantitative and semiquantitative features from medical images for improved decision support. It has the potential to uncover disease characteristics that are difficult to identify by viewing raw images alone. Given their advantages, researchers have explored the performance of radiomic features for chest X-ray analysis. For example, (Shi et al.; Saygılı, 2021) extracted a set of radiomic features to diagnose different types of pneumonia. (Bai et al., 2020) proposed a hybrid model to encode the combination of radiomic features and clinical information. (Ghosh et al., 2020) presented a new handcrafted feature to distinguish between severe and nonsevere patients. However, all the above methods rely on *accurate pathology localization annotations* to extract radiomic features from the correct "region of interest" (aka bounding boxes) but not other irrelevant parts(Van Griethuysen et al., 2017). Such bounding boxes are usually expensive and time-consuming to acquire by humans and, if inaccurate, will tremendously degrade the reliability of radiomic features. There is thus an unmet need to automatically localize pathologies on chest X-rays to facilitate radiomic features extraction.

Under the rapid development of deep learning, many researchers have made their efforts to utilize the Convolutional Neural Networks (CNNs) in building automated systems of chest X-ray abnormality classification and localization (Rajpurkar et al., 2017; Wang et al., 2017; Li et al., 2018; Liu et al., 2019b; Rozenberg et al., 2020; Wang et al., 2021). However, CNN methods witness several limitations.

First, chest X-rays own valuable domain knowledge and domain-specific features, such as radiomic features. Thus, they could have blessed better recognition but are unfortunately overlooked by most CNNs. Second, chest X-rays have more subtle discriminative features compared to natural images, making their recognition more challenging. Finally, CNNs are often criticized for being non-transparent and their predictions not traceable by humans, hence hinders their acceptance and adoption by clinicians.

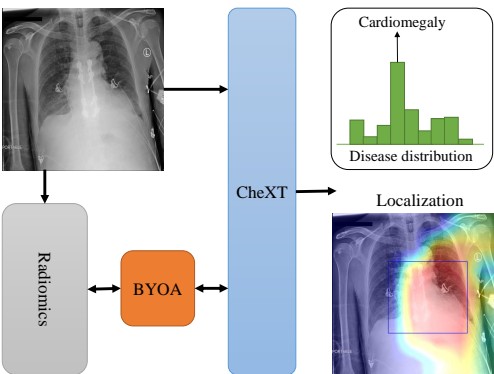

**Why Transformers for Chest X-rays?** The latest surge of transformers provides a promising alternative to model chest X-rays. Transformer was first prevailing to Natural Language Processing (Vaswani et al., 2017; Devlin et al., 2018; Brown et al., 2020), followed by its recent success in computer vision (Dosovitskiy et al., 2020; Carion et al., 2020; Zhu et al., 2020) and multi-modal learning (Ying et al., 2021). It is an "universal modeling tool" that can unify the feature extraction and fusion from different input modalities within one model, without domain-specific model tweaks. For example, (Akbari et al., 2021) demonstrated to learn powerful multi-modal representations from unlabeled video, audio, and text data, using one Transformer architecture.

Figure 1: General overview of our **CheXT** framework for Cardiopulmonary Disease Classification and Localization from chest X-rays. CheXT takes the chest X-ray image as the input and outputs a heatmap for pathology localization, based on which the bounding box could be obtained. Radiomic features are further extracted from the bounded region and are fed to predict the disease.

Bringing that into the context of chest X-rays, we see the tantalizing potential that *a Transformer could organically and jointly learn from two views of chest X-rays*: (i) raw X-ray images that contain the richest details, hence benefiting from the data-driven learning capacity; and (ii) radiomic features that encode critical domain prior knowledge, hence effectively guiding and regularizing the learning process. The appeal of Transformers is, however, blocked by a "chicken-and-egg" problem: as aforementioned, the extraction of reliable radiomic features hinge on the pathology localization, but then the pathology localization is often absent in images and also needs to be learned first.

This paper presents a holistic framework of *Knowledge-Guided Cross-Attention Transformer* for Chest X-ray analysis, named **CheXT** (Figure 1). CheXT consists of two Transformer-based branches that learn from two data formats characterizing the same patient: the image and radiomics branches. Both are deeply fused and interacted by cross-attention layers (Chen et al., 2021a). Notably, the radiomic features need be extracted from the learned pathology localizations, which are not readily available. The *key enabling technology* to resolve this hurdle, is to construct a "feedback loop" during training: the image branch leverages its learned attention to estimating pathology localization, which is then utilized to extract radiomic features from images in the radiomics branch. Training under a unified contrastive loss, such an end-to-end optimization loop can bootstrap accurate pathology localization from image data, with **no bounding boxes** used for training. Our contributions are outlined as follows:

- We leverage the radiomic feature as an "auxiliary input modality" correlated with the raw image modality and encoded with domain knowledge. We then propose a novel knowledge-guided cross-attention Transformer, **CheXT**, to jointly extract and fuse image and radiomic feature representations for chest X-ray analysis.

- To resolve the key "chicken-and-egg" problem of extracting radiomic features without available pathology localization, we construct an innovative optimization loop with the image and radiomic branches deeply interacting via attention. Such end-to-end loop can bootstrap accurate pathology localization from images without using human-annotated bounding boxes.

- Our approach achieve superior classification and localization results against several competitive baselines, on the NIH chest X-ray benchmark. Notably, compared to existing

approaches, CheXT generates more accurate disease localization for extracting radiomic features, by a significant average margin of 3.6% over different IoU thresholds.

## 2 RELATED WORK

**Radiomics in Medical Diagnosis.** The design of radiomics involves biological and medical data and prior knowledge. Thus, radiomics vastly enriches images and expands the horizons of the image toward in-vivo biologic information extraction (Gillies et al., 2016). In image-based biomarkers for cancer staging and prognostication, radiomics had shown promising power (Nasief et al., 2019). Radiomics extracts quantitative data from medical images to represent tumor phenotypes, such as spatial heterogeneity of a tumor and spatial response variations. (Eilaghi et al., 2017) demonstrated that radiomic of CT texture features are associated with the overall survival rate of pancreatic cancer. (Chen et al., 2017) revealed that the first-order radiomic features (e.g., mean, skewness, and kurtosis) are correlated with pathological responses to cancer treatment. (Huang et al., 2018) showed that radiomics could increase the positive predictive value and reduce the false-positive rate in lung cancer screening for small nodules compared with radiologists. (Zhang et al., 2018) found that multiparametric MRI-based radiomics nomograms provided improved prognostic ability in advanced nasopharyngeal carcinoma.

In comparison, deep learning is often criticized for being a "black box" and lacks interpretability despite high predictive accuracy. This limitation has motivated many interpretable learning techniques including activation maximization (Erhan et al., 2009), network inversion (Mahendran & Vedaldi, 2015), GradCAM (Selvaraju et al., 2016), and network dissection (Bau et al., 2017). We believe that the joint utilization of radiomics and interpretable learning techniques in our framework can further advance accurate yet interpretable learning in the medical image domain.

**Transformers for Medical Images.** Recently, Vision Transformer (ViT) (Dosovitskiy et al., 2020) achieved state-of-the-art classification on ImageNet by directly applying Transformers with global self-attention to full-sized images. Inspired by the promising performance of ViT, researchers have recently applied the idea to medical images. For example, (Oktay et al., 2018; Wang et al., 2019; Chen et al., 2021b) used the attention mechanism to boost the performance of medical image segmentation. (Valanarasu et al., 2021) proposed a gated axial-attention model to introduce an additional control mechanism in the self-attention module. (Park et al., 2021) utilized a hybrid framework of CNN and Transformer for Covid-19 prediction. However, those methods did not consider any domain prior knowledge. (Han et al., 2021) applied pre-extracted radiomic features to guide pneumonia detection from chest X-ray images. However, they adopted a convolutional backbone for image encoder, while using a another specifically crafted radiomic features encoder. Therefore, the method involves no joint interaction between image and radiomic features, and need to use accurate bounding boxes during training in order to extract radiomic features. Hence, their method dramatically limits usability in clinical practice.

## 3 METHOD

An overview of CheXT is illustrated in Figure 2. In the following subsections, we will first present Cross-attention Vision Transformer (CrossViT), a recent two-branch ViT backbone on which CheXT is built, and then describe our many unique improvements customized for Chest X-ray analysis.

### 3.1 PRELIMINARY: VIT AND CROSS-ATTENTION

ViT first converts an image into a sequence of patch tokens by dividing the image with certain patch size and linearly projecting each patch into tokens. A special token (CLS) is added in front of the sequence, as in the original BERT (Devlin et al., 2018). Then, all tokens are passed through stacked Transformer encoders. Finally, the hidden state corresponding to the CLS token is used as the aggregate sequence representation for image classification.

A Transformer encoder is composed of a sequence of blocks where each block contains multiheaded self-attention with a feed-forward network. Layer normalization and residual shortcuts are applied before and after every block, respectively. The granularity of the patch size affects the accuracy and complexity of ViT. Therefore, ViT can perform better with fine-grained patch size but with higher

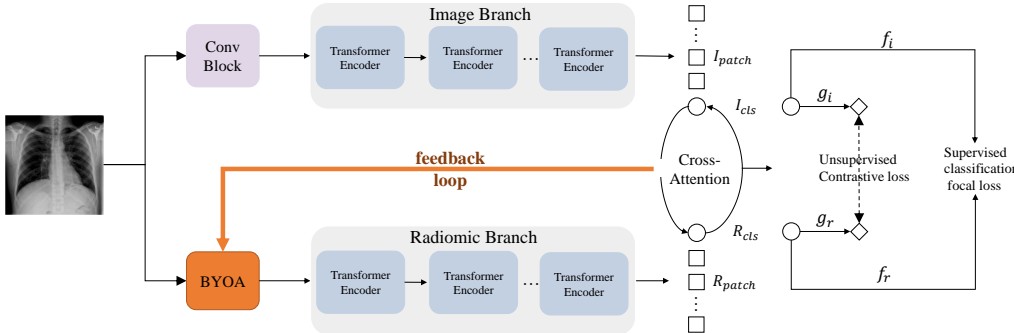

Figure 2: Overview of our proposed CheXT. It contains two branches, the Image branch and the Radiomics branch, to process the image and radiomic features (generated by the BYOA module shown in Figure 3), respectively. The output tokens are then fused by an efficient module via cross attention of the CLS tokens. Finally, the output of two CLS tokens ($I_{cls}$ and $R_{cls}$) are used for disease classification. We minimize the classification errors via a focal loss. Since we train with both labeled and unlabeled images, we leverage a contrastive learning strategy. Specifically, we generate the image view $z_i = g_i(I_{cls})$ by a projection head $g_i$. Similarly, we generate the radiomic view $z_r = g_r(R_{cls})$ by another projection head $g_r$. We maximize the agreement between $z_i$ and $z_r$ via a contrastive loss (NT-Xent). Of note, the contrastive loss is only active during the training.

FLOPS and memory consumption (Chen et al., 2021a). To relieve this problem, CrossViT (Chen et al., 2021a) was proposed with a dual-branch ViT where either branch operates at a different patch size, as its own "view" of the image. The cross-attention module is then used to fuse information between the branches to balance the patch sizes and complexity. Similar to ViT, the final hidden vector of CLS from two branches are used for image classification.

## 3.2 OUR PROPOSED CHEXT MODEL

CrossViT supplies a graceful framework to simultaneously tackle and fuse two different "views" from the same input data, e.g., different-size image patches in (Chen et al., 2021a). In CheXT, we extend their idea by treating image itself as one "view" and the radiomic feature extracted from the same image as another "view" (Figure 2). The two views are then jointly learned by interacting through cross-attention. Transformer serves as the modality-agnostic backbone for both.

Specifically, we introduce a dual-branch cross-attention Transformer where the first (primary) branch operates the image part, while the second (complementary) branch handles the radiomic features. To resolve the "chicken-and-egg" dilemma in extracting reliable radiomic features without bounding boxes, we have designed a novel _Bootstrap Your Own Attention_ (**BYOA**) module, using feedbacks to learn region localization for extracting radiomic features. A simple yet effective module is also utilized to fuse information between the branches. In the subsequent sections, we will describe the two branches, the BOYA module, and the fusion module.

**Image Branch.** The primary image branch uses a Progress-Sampling ViT (PS-ViT) (Yue et al., 2021) as its backbone. Unlike the vanilla ViT that splits images into fixed-size tokens, PS-ViT utilizes an iterative and progressive sampling strategy to locate discriminative regions and avoid over-partition object structures. We experimentally observed PS-ViT outperforms ViT and other variants in our framework because it generates higher-quality and more structure-aware attention maps, which are crucial for estimating the pathology localization during training.

**Radiomics Branch.** The complementary radiomics branch is for processing radiomic features. Handcrafted features usually cover a wide range of categories, such as first-order (basic intensity and shaped-based features), second-order (texture features extracted from various matrices), and more advanced features including those calculated from Fourier and wavelet transforms. Specifically, the radiomic features are composed of the following categories:

- First-Order statistics features to measure the distribution of voxel intensities within the bounding boxes. The features include energy (the measurement of the magnitude of voxel values),

Figure 3: Overview of our BYOA *module*. For the input chest X-rays, we look at the self-attention of the `CLS` token of the Image branch on the heads of the final output of the cross attention module. Then we apply a threshold (0.1) (which means we only keep the top 10% active pixels in the whole attention map) to generate the bounding boxes. Then with the generated bounding boxes, we use the *Pyradiomic* tool as the radiomic extractor to extract the radiomic features.

  entropy (the measurement of uncertainty in the image values), and max/mean/median gray level intensity within the region of interest.

- Shape-based features, such as Mesh Surface, Pixel Surface, and Perimeter.
- Gray-level features, such as Gray Level Co-occurrence Matrix (GLCM), Gray Level Size Zone (GLSZM), Gray Level Run Length Matrix (GLRLM), Neighboring Gray Tone Difference Matrix (NGTDM), and Gray Level Dependence Matrix (GLDM) features.

In short, radiomic features are a set of quantitative features that can describe the characteristics of medical images. In our framework, we aim to make the hidden features of the `CLS` similar to the radiomic features to learn the localization of pathologies in the chest X-rays. For this branch, we use the vanilla Transformer (Liu et al., 2017) as the radiomic features encoder. Please note that the only difference is that the positional encoding module is discarded, since there does not exist any positional relationship between the radiomic features.

**Bootstrap Your Own Attention (BYOA): A Feedback Loop Module.** Our main *roadblock* is how to generate robust radiomic features without pathology localization. On one hand, radiomic features are dependent on and highly sensitive to the local image regions of interest, for which we have no bounding box annotation. On the other hand, image features would benefit from the guidance from radiomic features that encode important domain knowledge. The learning of image and radiomic features are fully entangled, forming a challenging "chicken-and-egg" loop.

To address this issue, we design **BYOA** to constitute an end-to-end feedback loop that can bootstrap accurate pathology localization from image data, without any bounding box used for training (Figure 3). BYOA contains two components: attention map generation and radiomic feature extraction.

- *Attention Map Generation.* Similar to the approach in (Caron et al., 2021), we look at the self-attention of the `CLS` token on the heads of the last layer. Here, we have two `CLS` tokens from two branches, but the attention maps only come from the Image branch. Then we apply a threshold on the self-attention maps to generate bounding boxes for the extraction of radiomic features. The choice of the threshold will influence the quality of the radiomic features. Specifically, the threshold is designed as how much active pixels to keep in the generated attention map. The more active pixels we keep, the larger the generated bounding box is. Please see 4.3.3 for more details.
- *Radiomic Features Extraction.* Given the original images and generated bounding boxes, we used the Pyradiomic tool to extract radiomic features (Van Griethuysen et al., 2017).

**Cross-Attention Fusion Module.** This fusion involves the `CLS` token of one branch and patch tokens of the other branch. As the `CLS` token is the aggregate representation of the branch, this interaction helps include information at different scale. Please refer to (Chen et al., 2021a) for more details about the cross-attention mechanism.

### 3.3 SEMI-SUPERVISED LOSS FUNCTION

As shown in Figure 2, CheXT is trained using the linear combination of the supervised focal classification and unsupervised contrastive losses. For the supervised classification, considering that the chest X-ray dataset is usually highly imbalanced, we adopt the focal loss (Pasupa et al., 2020). For unsupervised contrastive learning, we use the cross-view contrastive loss (Chen et al., 2020).

**Supervised Classification Focal Loss.** We feed the output of the `CLS` tokens of two branches $I_{cls}$ and $R_{cls}$ to a simple linear classifier. The supervised classification focal loss $\mathcal{L}_{fl}$ is defined as

$$\mathcal{L}_{fl} = \begin{cases} -\alpha \left(1 - y'\right)^{\gamma} \log y', & y = 1 \\ -(1 - \alpha)y'^{\gamma} \log \left(1 - y'\right), & y = 0 \end{cases} \tag{1}$$

$\alpha$ allows us to give different importance to positive and negative examples. $\gamma$ is used to distinguish easy and hard samples and force the model to learn more from difficult examples.

**Unsupervised Cross-View Contrastive Loss.** Our contrastive loss extends the normalized temperature scaled cross-entropy loss (NT-Xent). The difference is that we maximize agreement between two feature views extracted from different input formats, one the image and the other radiomics.

Given an anchor chest X-ray in a minibatch, the positive sample will be its radiomic feature view, and the negative samples will be other chest X-rays (either image or radiomics). Since the `CLS` could be regarded as the representation of all other tokens, we only need to maximize the agreement between them. Suppose $I_{cls,k}$ and $R_{cls,k}$ are the $k-$th image features and radiomic features in the minibatch, respectively, and $sim()$ the cosine similarity, the loss function $\mathcal{L}_{cl}$ is defined as

$$\mathcal{L}_{cl} = -\log \frac{\exp(sim(g_i(I_{cls,k}), g_r(R_{cls,k}))/\tau)}{\sum_{k=1}^{N} \exp(sim(g_i(I_{cls,k}), g_r(R_{cls,k}))/\tau)} \tag{2}$$

where $\tau$ is the temperature. The final contrastive loss is summed over all instances in the minibatch.

Overall, we treat CheXT training as a semi-supervised multi-task learning. Since for chest x-rays, there exists two labels, disease class labels and pathology bounding box annotations. In our case, we only have access to the disease labels, however, bounding box annotations are more important than the disease class labels for chest x-rays. Here, when we say "semi-supervised", we refer to the bounding box annotations, not disease class labels. For multi-task, one task is supervised disease classification. The other is unsupervised cross-view contrastive learning. The total loss is:

$$\mathcal{L} = (1 - \lambda) \times \mathcal{L}_{cl} + \lambda \times \mathcal{L}_{fl} \tag{3}$$

## 4 EXPERIMENTS

### 4.1 DATASET AND PROTOCOL SETTING

The NIH Chest X-ray dataset (Wang et al., 2017) consists of 112,120 chest X-rays collected from 30,805 patients, and each image is labeled with 8 cardiopulmonary disease labels (Atelectasis, Cardiomegaly, Effusion, Infiltration, Mass, Nodule, Pneumonia, and Pneumothorax). The labels are extracted from the associated radiology report using an automatic labeler (Peng et al., 2018) with a reported accuracy of 90%. We use the extracted labels as ground-truth for training CheXT. The NIH dataset also includes high-quality bounding box annotations for 880 images by radiologists. We separated these 880 images from our entire dataset, and they are used only to evaluate disease localization. A significant difference between our method and existing baseline methods (Liu et al., 2019a; Li et al., 2018) is that our method does not require any training data related to the bounding box while others use some percentage of these images for training.

In our experiments, we followed the same protocol as in the studies of (Wang et al., 2017; Li et al., 2018), to shuffle our dataset (excluding images with Bounding Boxes annotations) into three subsets: 70% for training, 10% for validation, and 20% for testing. In order to prevent data leakage across patients, we make sure that there is no patient overlap between our train, validation, and test set.

### 4.2 IMPLEMENTATION DETAILS

We build our image branch encoder based on PS-ViT (Yue et al., 2021) and apply their default hyperparameters for training. The only difference is that we use a more shallow PS-ViT with 6 layers. For the radiomic branch encoder, since the radiomic features are already informative features, we use a more shallow Transformer (1-layers) to encode the radiomic features. And we add one more cross-attention layer to fuse these two format features. We set the batch size as 128 and train the model for 50 epochs (5 warm-up epochs). Other setup includes a cosine linear-rate scheduler with a linear warm-up, an initial learning rate of 0.004, and a weight decay of 0.05. During the evaluation, we resize the image to 256×256 and take the center crop 224×224 as the input.

| Method | Atelectasis | Cardiomegaly | Effusion | Infiltration | Mass | Nodule | Pneumonia | Pneumothorax | **Mean** |
|---|---|---|---|---|---|---|---|---|---|
| CNN | | | | | | | | | |
| (Wang et al., 2017) | 0.72 | 0.81 | 0.78 | 0.61 | 0.71 | 0.67 | 0.63 | 0.81 | 0.718 |
| (Wang et al., 2018) | 0.73 | 0.84 | 0.79 | 0.67 | 0.73 | 0.69 | 0.72 | 0.85 | 0.753 |
| (Yao et al., 2017) | 0.77 | 0.90 | 0.86 | 0.70 | 0.79 | 0.72 | 0.71 | 0.84 | 0.786 |
| (Rajpurkar et al., 2017) | 0.82 | 0.91 | **0.88** | 0.72 | 0.86 | 0.78 | 0.76 | **0.89** | 0.828 |
| (Kumar et al., 2018) | 0.76 | 0.91 | 0.86 | 0.69 | 0.75 | 0.67 | 0.72 | 0.86 | 0.778 |
| (Liu et al., 2019b) | 0.79 | 0.87 | **0.88** | 0.69 | 0.81 | 0.73 | 0.75 | **0.89** | 0.801 |
| (Seyyed et al., 2020) | 0.81 | **0.92** | 0.87 | 0.72 | 0.83 | 0.78 | 0.76 | 0.88 | 0.821 |
| (Han et al., 2020) | **0.83** | **0.92** | 0.87 | 0.76 | 0.85 | 0.76 | 0.77 | 0.86 | 0.828 |
| Transformer | | | | | | | | | |
| ViT | 0.74 | 0.78 | 0.81 | 0.72 | 0.70 | 0.66 | 0.65 | 0.76 | 0.728 |
| CrossViT | 0.69 | 0.71 | 0.72 | 0.72 | 0.74 | 0.79 | **0.82** | 0.88 | 0.759 |
| PS-ViT | 0.75 | 0.81 | 0.82 | 0.73 | 0.79 | 0.73 | 0.69 | 0.81 | 0.766 |
| CheXT | 0.80 | **0.92** | 0.78 | **0.86** | **0.88** | **0.88** | 0.79 | 0.81 | **0.839** |
| | ($\pm 0.02$) | ($\pm 0.00$) | ($\pm 0.01$) | ($\pm 0.01$) | ($\pm 0.02$) | ($\pm 0.00$) | ($\pm 0.01$) | ($\pm 0.02$) | – |

Table 1: Comparison with the baseline models for AUC of each class and average AUC. For each column, **bold** values denote the best results.

### 4.3 ABNORMALITY CLASSIFICATION

Abnormality classification is a multi-label classification problem. It assigns one or more labels among 8 cardiopulmonary diseases (Atelectasis, Cardiomegaly, Effusion, Infiltration, Mass, Nodule, Pneumonia, and Pneumothorax) to each input image at inference time. We conducted 3-fold cross-validation (Table 1). We compared CheXT with reference models, which have published state-of-the-art performance of disease classification on the NIH Chest X-ray dataset.

#### 4.3.1 EVALUATION METRIC

We used Area under the Receiver Operating Characteristics (AUC) to estimate the performance of our model (FawcettTom, 2006). A higher AUC score implies a better classification model. We also provide mean AUC across all the classes to highlight the overall performance of our model.

#### 4.3.2 COMPARISON WITH THE SOTA MODELS

AUC scores for each disease and mean AUC score across 8 diseases are presented in Table 1. Not only we compared CheXT with previous CNN-based SOTA models, but also several Transformer-based models. From the table, we can find that CheXT outperformed the baselines in the majority of diseases. We report the average AUC of 3 runs to show the robustness of our model. Compared to all baselines, CheXT achieves a mean AUC score of 0.839 using a dual-branch shallow Transformer across the 8 different diseases, which is 0.011 higher than the SOTA (uses DenseNet-121) (Rajpurkar et al., 2017) on disease classification. Specifically, our results significantly outperformed the best baseline models by 0.13, 0.09, and 0.02 in AUC for detecting Infiltration, Nodule, and Mass higher respectively. Besides, compared to the Transformer-based models, the key difference is that we utilize the extracted radiomic features for disease prediction, which improves the accuracy and adds the model's interpretability due to the utilization of self-designed handcrafted radiomic features.

#### 4.3.3 EFFECT OF DIFFERENT THRESHOLDS OF ATTENTION MAPS

We investigate the impact of the thresholds ($T$) in the process of attention map generation, on the performance of CheXT for disease classification. Figure 4A summarizes the AUC comparison of CheXT for different values of $T$. Higher values of $T$ imply smaller bounding boxes to extract radiomic features. During our experiments, we found that CheXT performs worse when very large bounding boxes are generated. This observation meets our expectations since radiomic features are very subtle to bounding boxes. Specifically, the extracted radiomic features are more robust if more focal and accurate bounding boxes are given.

#### 4.3.4 EFFECT OF CONTRASTIVE LEARNING

We also investigated the impact of contrastive learning. Specifically, we check the performance of CheXT for disease classification by varying $\lambda$ of equation (3). Figure 4B summarizes the AUC

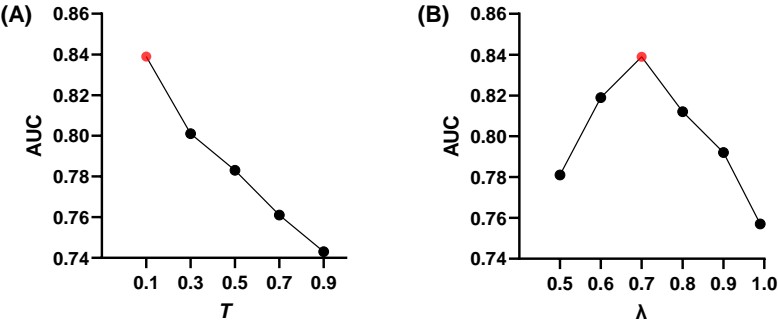

Figure 4: AUC comparison for varying $T$ in (A) attention map generation and (B) $\lambda$ in Equation 3.

comparison of CheXT for different values of $\lambda$. Higher values of $\lambda$ implies lower weight to contrastive loss. During our experiments, we found that CheXT performs worse when small weight (1%) is given to the contrastive loss. CheXT's performance improves when we increase contrastive loss weight, but after a certain point, it starts decreasing. This proves our hypothesis that both contrastive and focal losses are important. In a calculated ratio, they help CheXT to learn both disease-level and patient-level discriminative visual features.

## 4.4 ABNORMALITY LOCALIZATION

The NIH Chest X-ray dataset has 880 images labeled by radiologists with the bounding box information. We have used this dataset to evaluate the performance of CheXT for abnormality localization. Many prior works (Li et al., 2018; Liu et al., 2019a) have used a fraction of ground truth (GT) bounding boxes for training and evaluated their system on the remaining. To ensure a robust evaluation, we do not use any GT for training. Table 2 presents our evaluation results on all 880 images. We used (Wang et al., 2017) as our baseline to compare our localization results since it has the same experimental settings.

### 4.4.1 EVALUATION METRIC

For localization, we evaluated our detected regular rectangular regions against the annotated bounding boxes, using intersection over union ratio (IoU). Our localization results are only calculated for 880 images which have ground truth annotation for 8 diseases. The localization is defined as correct only if IoU > T(IoU). We evaluated CheXT for different thresholds ranging from {0.1, 0.2, 0.3, 0.4, 0.5, 0.6, 0.7} as shown in Table 2. A higher IoU threshold is preferred for disease localization because clinical usage requires high accuracy.

### 4.4.2 COMPARISON WITH THE SOTA MODELS

We compared disease localization accuracy under varying IoU, with baselines having similar settings as CheXT (Table 2). Unlike other baselines(Li et al., 2018; Liu et al., 2019a) that use a portion of 880 images for evaluation (because they need the remaining data for training), we used all 880 annotated images for evaluation. Therefore, no k-fold cross-validation for localization was performed. CheXT average performance across 8 diseases is significantly better than the baseline under all IoU thresholds. When the thresholds of IoU is set to 0.1, CheXT outperforms the baseline (Wang et al., 2017) in Cardiomegaly, Infiltration, Mass, and Pneumonia. Even with higher thresholds, our model performs superior to baseline. For example, when evaluated at T(IoU) = 0.5, our "Cardiomegaly" accuracy is still 32%, while the reference model achieves only 18%. Our "Pneumonia" accuracy is 12%, while the reference model has only 3% accuracy. Note that some diseases can appear at multiple places, but ground truth might have mentioned only one location. This can significantly impact the accuracy at high thresholds. More importantly, we also add the ViT as our additional baseline here, the results show that the radiomics branch and BYOA module can help the model learn more important regions, which also improves the model's interpretability. Few examples of the localization results of ViT and CheXT are shown in Figure 5.

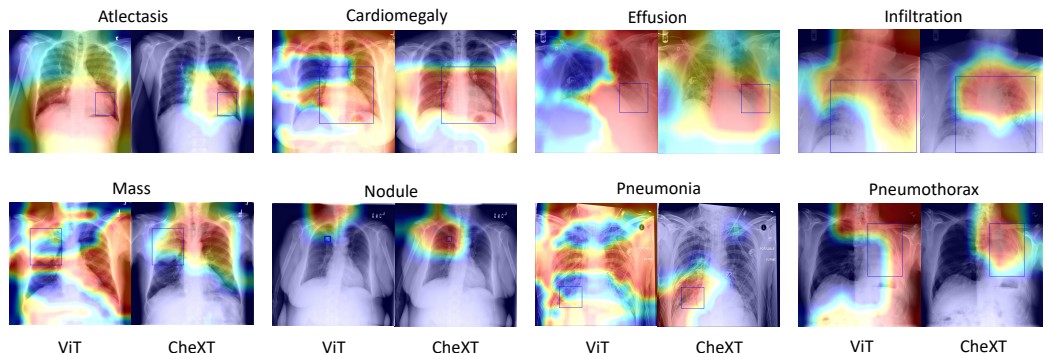

Figure 5: Examples of visualization of localization on the test images. The attention maps are generated from the self-attention maps of the `CLS` token. The ground-truth bounding boxes are shown in blue. The left image in each pair is the localization result of ViT (Dosovitskiy et al., 2020). The right one is our localization results. All examples are positive for corresponding disease labels. Best viewed in color.

| T(IoU) | Model | Atelectasis | Cardiomegaly | Effusion | Infiltration | Mass | Nodule | Pneumonia | Pneumothorax | **Mean** |
|---|---|---|---|---|---|---|---|---|---|---|
| 0.1 | (Wang et al., 2017) | 0.69 | 0.94 | 0.66 | 0.71 | 0.40 | 0.14 | 0.63 | 0.38 | 0.569 |
| | ViT | 0.58 | 0.91 | 0.61 | 0.77 | 0.44 | 0.11 | 0.75 | 0.25 | 0.553 |
| | CheXT | 0.61 | 0.95 | 0.65 | 0.82 | 0.50 | 0.13 | 0.79 | 0.28 | **0.591** |
| 0.2 | (Wang et al., 2017) | 0.47 | 0.68 | 0.45 | 0.48 | 0.26 | 0.05 | 0.35 | 0.23 | 0.371 |
| | ViT | 0.38 | 0.85 | 0.39 | 0.55 | 0.24 | 0.01 | 0.51 | 0.15 | 0.385 |
| | CheXT | 0.41 | 0.91 | 0.41 | 0.59 | 0.26 | 0.05 | 0.57 | 0.19 | **0.424** |
| 0.3 | (Wang et al., 2017) | 0.24 | 0.46 | 0.30 | 0.28 | 0.15 | 0.04 | 0.17 | 0.13 | 0.221 |
| | ViT | 0.20 | 0.45 | 0.19 | 0.32 | 0.06 | 0.00 | 0.21 | 0.02 | 0.181 |
| | CheXT | 0.28 | 0.79 | 0.22 | 0.38 | 0.12 | 0.01 | 0.41 | 0.05 | **0.283** |
| 0.4 | (Wang et al., 2017) | 0.09 | 0.28 | 0.20 | 0.12 | 0.07 | 0.01 | 0.08 | 0.07 | 0.115 |
| | ViT | 0.10 | 0.21 | 0.03 | 0.05 | 0.02 | 0.00 | 0.04 | 0.00 | 0.056 |
| | CheXT | 0.17 | 0.54 | 0.13 | 0.18 | 0.07 | 0.01 | 0.26 | 0.02 | **0.173** |
| 0.5 | (Wang et al., 2017) | 0.05 | 0.18 | 0.11 | 0.07 | 0.01 | 0.01 | 0.03 | 0.03 | 0.061 |
| | ViT | 0.05 | 0.15 | 0.01 | 0.04 | 0.02 | 0.00 | 0.03 | 0.00 | 0.034 |
| | CheXT | 0.08 | 0.32 | 0.05 | 0.09 | 0.05 | 0.00 | 0.12 | 0.01 | **0.090** |
| 0.6 | (Wang et al., 2017) | 0.02 | 0.08 | 0.05 | 0.02 | 0.00 | 0.01 | 0.02 | 0.03 | 0.029 |
| | ViT | 0.01 | 0.03 | 0.01 | 0.01 | 0.01 | 0.00 | 0.01 | 0.00 | 0.010 |
| | CheXT | 0.02 | 0.15 | 0.03 | 0.04 | 0.03 | 0.00 | 0.06 | 0.00 | **0.041** |
| 0.7 | (Wang et al., 2017) | 0.01 | 0.03 | 0.02 | 0.00 | 0.00 | 0.00 | 0.01 | 0.02 | 0.011 |
| | ViT | 0.00 | 0.00 | 0.00 | 0.01 | 0.00 | 0.00 | 0.00 | 0.00 | 0.001 |
| | CheXT | 0.01 | 0.04 | 0.01 | 0.02 | 0.01 | 0.00 | 0.03 | 0.00 | **0.015** |

Table 2: Disease localization under varying IoU on the NIH Chest X-ray dataset. Please note that since our model doesn't use any ground truth bounding box information, to fairly evaluate the performance of our model, we only consider the previous methods' results under the same setting,

## 5 CONCLUSION

In this paper, we propose an end-to-end knowledge-guided cross-attention Transformer, named CheXT that can jointly model abnormality classification and localization without the supervision of localization annotation of chest X-rays. Our approach differs from previous studies in the choice of universal modeling transformer, the use of radiomic features as prior knowledge, and a feedback loop for image and radiomic features to mutually interact with each other. Additionally, the project aims to mitigate current gaps in radiology by making prior knowledge more accessible to image data analytic and diagnostic assisting tools, with the hope that this will increase the model's interpretability. Experimental results demonstrate that our method outperforms the state-of-the-art algorithms, especially for the disease localization task, where our method can generate more accurate bounding boxes.

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
