# OpenReview forum: "CheXT: Knowledge-Guided Cross-Attention Transformer for Abnormality Classification and Localization in Chest X-rays"
_ICLR.cc/2022/Conference — ICLR 2022 Submitted_

### Official Review · Reviewer_ZMa3 · 2021-11-01

**Correctness:** 3
**Technical Novelty And Significance:** 3
**Empirical Novelty And Significance:** 3
**Recommendation:** 5
**Confidence:** 5

**Main Review:**

Strengths
+ The paper is fairly well written and presents a well-studied literature review.
+ The paper proposes an end-to-end transformer-based disease classification and localization model incorporating radiomic feature knowledge with cross-attention. The idea of bounding box generation and radiomic feature extraction in a feedback loop module is new and interesting.
+ Experimental setting including the ablation study for the effect of thresholds on attention maps and the effect of contrastive learning is good.

Weaknesses
- Exposition of results is not good. CheXT is compared against some weak baselines. A more appropriate CNN baseline should be something that includes similar radiomic feature extraction and bounding box generation [1]. Also, [1] is not cited in Related Work.
- The paper claims significant performance improvement by the CheXT model although no evidence (statistical test) is provided to support the claim.
-  Classification performance drops for some of the disease classes (effusion, pneumonia, pneumothorax) after incorporating the radiomics branch. It looks like radiomics branch is hurting the performance for these classes. Is there any impact of label distribution?
- In Fig. 3, two images look different. Why so?
- The training of CheXT is not necessarily semi-supervised as every image has at least disease class label.
- It is not clear how to obtain f_i and f_r; y, y' are not defined Eqn. (1).
- It would be more viable to report both classification and localization performances on the same 880 images. Is there any correlation between e.g., classification AUC and localization IoU?
- It is not clear how the model could improve generalization. There is no generalization assessment of the model for out-of-distribution cases. For example, using other datasets such as, CheXpert and MIMIC-CXR.
- In 4.1, the paper mentions the train/test/val splits, but Table 1 reports results after 3-fold cross-validation. They don't look consistent.
- In 4.4.2: "when the IoU is set to 0.1" should be threshold of IoU. "For Cardiomegaly, Infiltration, and Pneumonia." incomplete sentence.
- It is not clear how higher values of \lambda implies lower weight to contrastive loss in Eqn. (3). How does the model perform without the contrastive loss at all (\lambda=0)?
- There is no mention of the execution speed, computing resource, and failure cases (if any) for the proposed model.
- There are certainly other Transformer-based chest X-ray models such as [2].

1. https://arxiv.org/abs/2011.12506
2. https://arxiv.org/abs/2104.07235

**Summary Of The Paper:**

This paper presents a Transformer-based model CheXT for abnormality classification and localization from chest X-rays with auxiliary modality of radiomic features via a feedback loop module. The output tokens from image and radiomics branches are fused by a cross-attention module for the localization and the disease classification is performed from the output of two CLS tokens. The model is trained by jointly optimizing supervised classification focal loss and unsupervised cross-view contrastive loss. The evaluation on the NIH chest X-ray dataset for 8 cardiopulmonary diseases shows improved performance for disease classification and localization.

**Summary Of The Review:**

This paper presents a Transformer-based model for joint classification and localization of diseases from chest X-ray images, leveraging an image branch and a radiomics branch via a feedback loop module of bounding box generation. While the proposed method has novel and interesting components, the experimentation and evaluation in the paper need to be improved. The proposed CheXT should be compared against the more recent CNN baselines such as ChexRadiNet [1]. There are also some concerns with the experimental details and analyses that need to be addressed.

---

> ### Author Response · Authors · 2021-11-20
> **Response to Reviewer ZMa3**
>
> Thanks for acknowledging the novelty of incorporating radiomic features for chest x-rays analysis. We provide the detailed responses to your comments point by point below.
>
> $\textbf{Q1}$. Where is the statistical test?
>
> For our CheXT model, we ran it 3 times with different random seeds, and provided the standard error. For the other baselines, we collected the results from the reported values by authors. And they do not provide statistical test numbers.
>
> $\textbf{Q2}$. Why does performance drop for some diseases after incorporating radiomic features?
>
> Radiomic features highly rely on the bounding boxes annotations. Some diseases (e.g., effusion, pneumonia and pneumothorax) are located in the whole lung area, which is less discriminative for radiomic features. Other diseases, like Mass and Nodule, are more focal. Therefore their performance improves significantly after incorporating radiomic features.
>
> $\textbf{Q3}$. In Fig3, two images look different, why so?
>
> Thanks for pointing this out. This is a typo. We will fix this figure.
>
> $\textbf{Q4}$. Is CheXT semi-supervised?
>
> Fairly speaking, our terminology is not clear here. Since for chest x-rays, there exists two labels, disease class labels and pathology bounding box annotations. In our case, we only have access to the disease labels, however, bounding box annotations are more important than the disease class labels for chest x-rays. Here, when we say "semi-supervised", we refer to the bounding box annotations, not disease class labels, we will clarify it in paper.
>
> $\textbf{Q5}$. How to obtain f_i and f_r, and what is y, y’?
>
> The f_i and f_r are two different MLP networks for the final classification. y and y’ represent the ground truth disease label and predicted disease label, respectively.
>
> $\textbf{Q6}$. Train/val/test split and 3-fold cross validation inconsistent, incomplete sentences, unclear lambda and how does the model perform without contrastive loss?
>
> Many thanks for pointing this out. Actually, all the above are typos. We used the public data split and evaluated our model three times. A higher lambda implies a lower weight to contrastive loss ( another typo in Eq. 3). lambda = 1 means no contrastive loss. We will fix all the typos in the final version.
>
> $\textbf{Q7}$. How is the execution speed, computing resource and failure cases for CheXT?
>
> Compared with other methods, our method requires extra time to extract radiomic features which should be insignificant as compared to the time of training. For the computing resources, we use 8 Nvidia V100 16GB GPUs from AWS. The obvious failure cases that we observed are Nodule and Mass detection because the bounding boxes of these two diseases are usually too small to detect.
>
> $\textbf{Q8}$. More datasets, like CheXpert and MIMIC?
>
> Both CheXpert and MIMIC have no bounding box annotations to evaluate the disease localization.
>
> $\textbf{Q9}$. Classification performance of 880 images? Is there any correlation between disease classification and localization?
>
> The 880 images are already contained in the test set. However, we agree that it is more viable to separately report the classification performance of these 880 images. We will report it in the final version. In general, the higher disease classification implies more accurate localization.
>
> $\textbf{Q10}$. Other Transformer-based related works.
>
> We will add these works to our related work in the final version.

---

> > ### Author Response · Authors · 2021-11-27
> > **Follow-up and a Kind Reminder**
> >
> > Dear Reviewer ZMa3,
> >
> > We thank the reviewer time for the review, and we hope to have a further discussion with reviewer ZMa3 to see if our response solves the concerns.
> >
> > We would sincerely appreciate it if reviewer ZMa3 could reply to the most important points in our rebuttal. For example, we discussed the reasoning behind the performance drop for some diseases after incorporating radiomic features. We have also improved the overall presentation of the paper and have addressed all typos and confusing parts you mentioned. Please kindly have a check on the updated PDF.
> >
> > We genuinely hope reviewer ZMa3 could kindly check our response. And we are more than willing to answer additional questions and provide more information or clarification, should it be necessary. Thank you!
> >
> > Best wishes,
> >
> > Authors

---

### Official Review · Reviewer_LbVM · 2021-11-01

**Correctness:** 3
**Technical Novelty And Significance:** 2
**Empirical Novelty And Significance:** 2
**Recommendation:** 5
**Confidence:** 4

**Main Review:**

1) Radiomics is a term that most people in the ICLR community are not familiar with. What is it exactily? Is it a set of radiological/image features or bio-markers in medical images? Or something else? The authors should consider explaining it in plain language or ML language.

2) Prior multi-modal feature fusion work in the chest x-ray domain should be discussed, examples below.

    - Moradi, Mehdi, Ali Madani, Yaniv Gur, Yufan Guo, and Tanveer Syeda-Mahmood. "Bimodal network architectures for automatic generation of image annotation from text." arXiv preprint arXiv:1809.01610 (2018).

    - Wang, Xiaosong, Yifan Peng, Le Lu, Zhiyong Lu, and Ronald M. Summers. "TieNet: Text-Image Embedding Network for Common Thorax Disease Classification and Reporting in Chest X-rays." arXiv e-prints (2018): arXiv-1801.

    - Xue, Yuan, and Xiaolei Huang. "Improved disease classification in chest x-rays with transferred features from report generation." In International Conference on Information Processing in Medical Imaging, pp. 125-138. Springer, Cham, 2019.

    - Liao, Ruizhi, Daniel Moyer, Miriam Cha, Keegan Quigley, Seth Berkowitz, Steven Horng, Polina Golland, and William M. Wells. "Multimodal Representation Learning via Maximization of Local Mutual Information." arXiv preprint arXiv:2103.04537 (2021).

3) I don't see "prior knowledge" and how it's utilized in this work. Isn't the so called prior knowledge in this paper essentially extracted from the data (chest x-ray images)?

4) What are the patch sizes? How are they determined?

5) There are 14 pathology labels in the NIH chest x-ray dataset. Why/how did the authors select the 8 classes out of the 14?

6) In Table 1, make the best results bold instead of in red.

7) Are the AUC values generated on the test set? If so, I'm concerned about overfitting by over tuning the hyper-parameters.

**Summary Of The Paper:**

This paper has proposed a machine learning model to utilize radiomics features in order to improve chest x-ray image classification performance.

**Summary Of The Review:**

Overall this paper is well written besides several comments I made above. The classification results on AUC are not convincing to me that integrating radiomics features in this proposed way is better than the state of the art.

---

> ### Author Response · Authors · 2021-11-20
> **Response to Reviewer LbVM**
>
> Thank you for reviewing the paper and providing your valuable feedback. We provide pointwise detailed responses to your concerns, hoping them to help to build up further understanding.
>
> $\textbf{Q1}$. What is the difference between CheXT with previous multi-modal feature fusion work?
>
> Both CheXT and multi-modal feature fusion works use two branches, but their motivation and implementation are very different.
>
> CheXT is a single-modal framework (image) that implicitly incorporates radiomics as prior knowledge. Differently, the multi-modal feature fusion works require two separate modals (e.g., text and image). The challenging part of our study is radiomics extraction which needs bounding box annotations that are not available in the chest x-rays dataset. To solve this problem, we propose a new self-bootstrapping strategy ($\textbf{BYOA}$) to localize bounding boxes via an attention-based feedback loop. Images and radiomics branches hence become highly interactive and reinforce each other. We showed that such a self-bootstrapping strategy achieves better and more interpretable performance.
>
> $\textbf{Q2}$. What are the patch sizes? How are they determined?
>
> We do not search for the optimal patch size. Instead, we set the patch size to 16x16 because most diseases’ areas in the chest x-ray are greater than 16x16.
>
> $\textbf{Q3}$. Are the AUC values generated on the test set? If so, I'm concerned about overfitting by over tuning the hyper-parameters.
>
> Yes. NIH chest x-ray dataset has a public data split. We follow the same experimental settings in previous work in our study. We do not observe overfitting issue based on our experiments.

---

> > ### Author Response · Authors · 2021-11-27
> > **Follow-up and a Kind Reminder**
> >
> > Dear Reviewer LbVM,
> >
> > We thank the reviewer time for the review, and we hope to have a further discussion with reviewer LbVM to see if our response solves the concerns.
> >
> > We would sincerely appreciate it if reviewer LbVM could reply to the most important point in our rebuttal. Specifically, we discussed that our CheXT model and previous multi-modal feature fusion works are very different in their motivation and implementation.
> >
> > We genuinely hope reviewer LbVM could kindly check our response. And we are more than willing to answer additional questions and provide more information or clarification, should it be necessary. Thank you!
> >
> > Best wishes,
> >
> > Authors

---

### Official Review · Reviewer_euAn · 2021-11-02

**Correctness:** 3
**Technical Novelty And Significance:** 2
**Empirical Novelty And Significance:** 2
**Recommendation:** 5
**Confidence:** 4

**Main Review:**

Strength:
- The proposed approach and considering radiomic features along with learnt features is valuable and very relevant to the community.

Main Concerns and comments to improve the paper:
- The paper does not consider evaluation of the method on more publicly available datasets and the study is only limited to NIH-14 Chest X-ray data.
- Baselines and state of the art results on the test data: One important baseline is to make the prediction using only radiomic features.  Also,  [Han’20]  has reported much higher performance on NIH-14 and for the same task which they have not been included in the paper.
Improvement over the CNN baseline on classification: We see 0.011 improvement in mean AUC over Rajpurkar’17 as one of the baseline which is not significant.
- Details of the comparison setups are missing. I am wondering if these numbers are collected from the reported values by authors or they ran every SOTA against their data division (less likely since there is no error bound reported for SOTAs). And if they are-implemented or re-evaluated each method what are the details and consideration for a fair comparison and reproducing the results.
- Main contribution of the paper is to activate usage or radiomic features within the blackbox  structure of deep networks, however, there is no discussion or insight around the meaning and importance of multiple radiomic features that have been used. There are also works that include prior knowledge for Chest X-ray classification including [Han’20]. These features either unlock a better performance (which seems not being really successful in) or demystify the learnt feature space in a way that is pleasant for human interpreters. The paper does not provide such results.


Suggestion to improve the work:
- It would be useful to add some more explanation around radiomics term, feature extraction procedure and meaning of it, since this is not a common term for the machine learning community.
- Consider boosting the results and observation using a few more test data such as MIMIC and CheXpert that have the similar label space as NIH-14 and publicly being used in most of the related papers.
- The above mentioned datasets also can be used for evaluation of the out-of-domain generalization of the method which has not been discussed in the paper.
- Experimental setup and evaluation toward the SOTA can improve significantly. For the comparison purpose I would consider producing a statistical error range with either k-fold cross validation or boot-strapping for both CheXT and SOTAs. I would personally see a proper statistical comparison between a single best SOTA with the proposed method more valuable than having multiple models without any statistical validation.
- Adding interpretation around radiomic feature meaning, how the proposed method unlocked one or more of the 3 listed limitations of CNN methods that has been discussed in the Introduction.
- It is worth also discussing failure cases and the reasoning behind lower performance for some of the conditions such as effusion and pneumonia.

[Han'20] Han, Yan, Chongyan Chen, Liyan Tang, Mingquan Lin, Ajay Jaiswal, Song Wang, Ahmed Tewfik, George Shih, Ying Ding, and Yifan Peng. "Using Radiomics as Prior Knowledge for Thorax Disease Classification and Localization in Chest X-rays." arXiv preprint arXiv:2011.12506 (2020).

**Summary Of The Paper:**

The paper presents a transformer-based architecture for joint localization and classification (of 8 different conditions) of the Chest X-ray images in NIH dataset. The main contribution of this paper is utilization of prior knowledge and radiomic features that have conventionally been used for Chest X-ray condition detection including first order statistics, second-order statistics, shape-based features, and etc.

**Summary Of The Review:**

The main contribution of the paper is enabling the deployment of radiomic features alongside ViT architecture for prediction of Chest abnormality. The study is somehow limited to only one dataset, the results are not supporting the claims and limited. There is no insight provided about how this new feature helps with current shortcomings of CNN including explainability.

---

> ### Author Response · Authors · 2021-11-20
> **Response to Reviewer euAn**
>
> Thank you for acknowledging the value of utilizing the radiomic features along with learnt features for chest X-rays analysis. We appreciate your comments to help make our paper stronger. We provide detailed responses to your concerns point by point below.
>
> $\textbf{Q1}$. How is the performance of using only radiomic features?
>
> Thanks for pointing this out.  We performed disease classification using only radiomic features, and obtained an average AUC of 0.815 on 8 diseases. The detailed results are shown below.
>
> |Method|Atelectasis|Cardiomegaly|Effusion|Infiltration|Mass|Nodule|Pneumonia|Pneumothorax|Mean|
> | :--- | :---: | :---: | :---: | :---: | :---: | :---: | :---: | :---: | :---: |
> |Radiomic features|0.82|0.90|0.86|0.71|0.85|0.76|0.74|0.88|0.815|
> |CheXT|0.80|0.92|0.78|0.86|0.88|0.88|0.79|0.81|0.839|
>
> $\textbf{Q2}$. Details about comparison setup?
>
> We appreciate your comments about the comparison setup. Since the NIH chest x-ray dataset has a public data split, all baselines followed the same data split method. We collect the results from the reported values by authors.
>
> $\textbf{Q3}$. How could radiomic features unlock the limitations of CNN methods?
>
> We would clarify our discussion, but we want to respectfully argue that our method achieved better and more strongly interpretable performance.
>
> Compared with state-of-the-art CNN methods in NIH chest x-ray datasets, our method improves the average AUC of 8 diseases by 1.1%. In addition, our methods can also provide more accurate bounding boxes prediction, which is usually ignored by previous studies. For interpretability, instead of black-box feature learning from chest x-rays for disease classification, we demystify this opaque process into pathology localization, radiomic features extraction and classification process. This process is more interpretable than previous studies since the radiomic features involves biological and medical knowledge that is
>  acceptable by clinicians.

---

> > ### Author Response · Authors · 2021-11-27
> > **Follow-up and a Kind Reminder**
> >
> > Dear Reviewer euAn,
> >
> > We thank the reviewer time for the review, and we hope to have a further discussion with reviewer euAn to see if our response solves the concerns.
> >
> > We would sincerely appreciate it if reviewer euAn could reply to the most important points in our rebuttal. For example, we have added additional baselines of [Han’20] and using only radiomic features, all reported numbers positively show clear performance improvements, which indicates the benefits of incorporating radiomic features for chest x-ray analysis. Besides, we have discussed why radiomic features could help unlock the limitations of CNN methods. And we have discussed the failure cases and the reasoning in the response to reviewer ZMa3.
> >
> > We genuinely hope reviewer euAn could kindly check our response. And we are more than willing to answer additional questions and provide more information or clarification, should it be necessary. Thank you!
> >
> > Best wishes,
> >
> > Authors

---

### Author Response · Authors · 2021-11-20
**General Respons to All Reviewers**

We thank the reviewers for the constructive feedback and valuable comments. Here we address the concerns.

$\textbf{Q1}$. More explanation about radiomic features?

In medicine, radiomics is a method to extract a large number of features from medical images using data-characterization algorithms. These features, termed radiomic features, can uncover tumoral patterns and characteristics that fail to be appreciated by the naked eye. The extraction of radiomics has two major steps, image segmentation and feature extraction. First, image segmentation will segment the images to essential parts, in this case, the disease of Chest X-rays, which are called "region of interest". Then, many interpretable size and shape-based features can be computed for the "region of interest". Therefore, radiomic features are highly related to the "region of interest".

$\textbf{Q2}$. Why are radiomic features considered as additional prior knowledge?

The design of radiomics involves biological and medical data that will improve diagnostic accuracy and predictive power for decision support. The radiomic features thus implicitly incorporate medical knowledge and go beyond traditional image features extracted from images.

In this study, we demonstrate that  radiomics tremendously enriches images and expands the horizons of the image toward extraction of in vivo biologic information. Therefore, we think radiomics is significantly different from conventional features [1] and can be treated as prior auxiliary knowledge.

$\textbf{Q3}$. Why do we only select the 8 diseases rather than 14 in the NIH chest x-ray dataset?

The 14 diseases in the NIH chest x-ray can be categorized into two classes, 8 focal diseases (selected and used in our method) and 6 diffuse diseases. The focal disease is located at a specific and distinct area such as the heart (for cardiomegaly) and lung (for pneumothorax). The diffuse disease involves all lobes of both lungs, as opposed to in just one spot (like pneumothorax). Therefore, we believe it is clinically not appropriate to predict bounding boxes for diffuse diseases. In this study, we select the 8 focal diseases out of 14 diseases in the NIH chest x-ray dataset.

$\textbf{Q4}$. One missing better baseline [2]?

Thanks for pointing this out. We re-trained the model in [2] on 8 focal diseases. The results are shown below. We will include the new results in the final version.

|Model|Atelectasis|Cardiomegaly|Effusion|Infiltration|Mass|Nodule|Pneumonia|Pneumothorax|Mean|
| :---- | :----: | :----: | :----: | :----: | :----: | :----: | :----: | :----: | :----: |
|ChxRadiNet[2]|0.83|0.92|0.87|0.76|0.85|0.76|0.77|0.86|0.828|
|CheXT|0.80|0.92|0.78|0.86|0.88|0.88|0.79|0.81|0.839|

[1] Gillies, Robert J., Paul E. Kinahan, and Hedvig Hricak. "Radiomics: images are more than pictures, they are data." Radiology 278, no. 2 (2016): 563-577.

[2] Han, Yan, Chongyan Chen, Liyan Tang, Mingquan Lin, Ajay Jaiswal, Song Wang, Ahmed Tewfik, George Shih, Ying Ding, and Yifan Peng. "Using Radiomics as Prior Knowledge for Thorax Disease Classification and Localization in Chest X-rays." arXiv preprint arXiv:2011.12506 (2020).

---

### Author Response · Authors · 2021-11-23
**Follow-up and a Kind Reminder**

Dear Reviewers,

We want to thank all the reviewers for your constructive feedback and valuable comments, again. As a follow-up to our rebuttal, we want to kindly remind you that the discussion period will end soon. We hope to utilize this open discussion period to discuss our paper more, answer additional questions, and improve the quality of our submission. Have you gotten a chance to read the below responses, which attempt to address all your concerns?

Specifically, as Reviewer euAn and LbVM suggested, we have added more explanations of radiomic features and why it could be regarded as additional prior knowledge. And we have compared with the additional baseline [Han'20] suggested by Reviewer euAn and ZMa3. And regarding the Reviewer ZMa3's concerns on typos, we have fixed them and have uploaded the latest PDF.

We sincerely hope to have more further discussions with you to see if our response solves the concerns. We would be more than willing to answer additional questions or provide more information, should it be necessary, and hope our paper could receive a positive and fair assessment.

Best,

Paper 1843 Authors.

---

### Decision · Program_Chairs · 2022-01-20

**Decision:**

Reject

**Comment:**

This paper is on the right track to be accepted after a revision but it is not ready yet. The reviewers were mostly puzzled by different details in the evaluation process, however, as far I can see, most of them should be possible to address. My impression is also that the authors might be more used to a slightly different style of paper writing, more similar to what is typically accepted at MICCAI, ML4H, MIDL etc. I think that a lot can be improved in this respect just by carefully analyzing the reviews.